# ROBUSTNESS VIA LEARNED BREGMAN DIVERGENCE

## ABSTRACT

We exploit the Bregman divergence to generate functions that are trained to measure the semantic similarity between images under corruptions and use these functions as alternatives to the $L^p$ norms to define robustness threat models. Then we replace the projected gradient descent (PGD) by semantic attacks, which are instantiations of the mirror descent, the optimization framework associated with the Bregman divergence. Adversarial training under these settings yield classification models that are more robust to common image corruptions. Particularly, for the contrast corruption that was found problematic in prior work we achieve an accuracy that exceeds the $L^p$- and the LPIPS-based adversarially trained neural networks by a margin of 29% on the CIFAR-10-C corruption dataset.

## 1 INTRODUCTION

Neural networks for image classification are sensitive to input variations the way the human vision system is not. This means they can misclassify when images are subjected to either small maliciously-crafted perturbations (so-called adversarial examples) (Biggio et al., 2013; Szegedy et al., 2014; Papernot et al., 2016), or to the more realistic distribution shifts associated with common, realistic image corruptions like blur or contrast changes (Dodge & Karam, 2017; Hendrycks & Dietterich, 2019). Thus both adversarial robustness and corruption robustness are active research topics.

The common approach to achieve corruption robustness is the augmentation of the training set with synthetically modified images, e.g., mixing images by blending, cutting and pasting parts, fusion through spectral analysis, and others (Zhang et al., 2018a; Yun et al., 2019; Harris et al., 2021; Walawalkar et al., 2020; Park et al., 2022; Yin et al., 2022; Liang et al., 2023).

The most successful method for achieving adversarial robustness is adversarial training (AT) (Madry et al., 2018) and its many follow-up variants, e.g., (Uesato et al., 2019; Zhang et al., 2019; Carmon et al., 2019; Wu et al., 2020; Chen et al., 2021; Rebuffi et al., 2021; Jiang et al., 2023). It consists of first performing an attack to find adversarial examples images within an $\epsilon$-ball w.r.t. an $L^p$ norm, and then including these in the training set. The basic machinery for most white-box attacks is projected gradient descent (PGD) (Madry et al., 2018; Goodfellow et al., 2015; Wong et al., 2020; Croce & Hein, 2020). Even though $L^p$ norms carry no semantic meaning (they are oblivious to the nature of images relevant to humans), AT was found to also improve corruption robustness when done with carefully chosen hyperparameters (Hendrycks & Dietterich, 2019; Ford et al., 2019; Xie et al., 2020; Kang et al., 2019; Kireev et al., 2022). Conversely, Ford et al. (2019) proved that adversarial examples exist due to a nonzero test error under random noise, a particular distribution shift. Thus, the hope is to further improve corruption robustness by using the AT machinery with more meaningful similarity measures. A prominent example is the work by Kireev et al. (2022); Laidlaw et al. (2021); Wang et al. (2021). It uses the so-called learned perceptual image patch similarity (LPIPS) (Zhang et al., 2018b), which computes the $L^2$ form over the features extracted by a convolutional neural net, and ports PGD accordingly.

In this work, we take a different route by first learning a *Bregman divergence* (Bregman, 1967) for image corruptions and then using the associated *mirror descent* optimization framework (Nemirovskij & Yudin, 1983), which generalizes PGD, for AT.

**Bregman divergence and mirror descent.** Bregman divergence (Bregman, 1967) is a generalization of the Kullback–Leibler divergence (Kullback & Leibler, 1951), and is widely used in statistics and information theory to define distances in spaces where the Euclidean geometry is not appropriate

such as probability distributions, covariance descriptors, random processes and others (Chowdhury et al., 2023; Csiszar & Matus, 2008; Bauschke & Borwein, 1997; Stummer & Vajda, 2009; Frigyik et al., 2008; Harandi et al., 2014). Once defined, the associated mirror descent (Nemirovskij & Yudin, 1983), which generalizes PGD, allows solving constrained optimization problems including, as we will show, the kind needed in adversarial attacks and training.

**Contributions.** In this paper we offer progress in the quest for corruption robustness through a theoretically principled approach that uses a learned Bregman divergence with a suitably designed AT. A Bregman divergence is defined by a so-called base function that is convex and has an invertible gradient. As example, the KL divergence is defined by the Shannon entropy.

For a given corruption type we show how to learn an eligible base function as a neural net using a proposed self-supervised algorithm. The associated Bregman divergence is semantic in that it assesses corrupted images as close and randomly perturbed ones as far from the clean image, even if in Euclidean distance it is clearly the opposite.

We instantiate mirror descent, which generalizes PGD, to perform semantic adversarial attacks using the learned divergence instead of using an $L^p$ norm.

We adopt this attack for AT and show that it yields classification models with high corruption robustness on the CIFAR-10-C for the contrast and fog corruptions that are known to be problematic (e.g., Ford et al. (2019) and Kireev et al. (2022)).

## 2  BACKGROUND

We first recall standard adversarial training (AT) with projected gradient descent (PGD). Then we provide background on Bregman divergence (Bregman, 1967) and the associated mirror descent framework, which generalizes PGD (Nemirovskij & Yudin, 1983). Our work will then port AT with PGD using an $L^p$ norm to one with mirror descent and a learned Bregman divergence as similarity measure.

**Adversarial training.** Let $l(\boldsymbol{x}, y; \theta)$ be a loss of a classifier parameterized by $\theta$ where the input image $\boldsymbol{x}$ and the label $y$ are sampled from the data distribution $\mathcal{D}$. As formalized by Madry et al. (2018), training an adversarially robust model amounts to solving the following min-max optimization problem:

$$\min_{\theta} \mathbb{E}_{(\boldsymbol{x}, y) \sim \mathcal{D}} \left[ \max_{\boldsymbol{x}' \in \mathbb{S}(\boldsymbol{x})} l(\boldsymbol{x}', y; \theta) \right] \tag{1}$$

where $\mathbb{S}(\boldsymbol{x})$ is the set of images that are considered similar to $\boldsymbol{x}$. Under the common $L^p$ threat model, $\mathbb{S}(\boldsymbol{x})$ is defined as an $L^p$ ball centered on $\boldsymbol{x}$ of chosen radius $\epsilon$: $\mathbb{S}(\boldsymbol{x}) = \mathbb{B}(\boldsymbol{x}, \epsilon)$[1]. In this case, the inner maximization problem is solved by PGD, which consists of iterating over two steps: a gradient-based update followed by a projection into $\mathbb{B}(\boldsymbol{x}, \epsilon)$.

**Bregman divergence.** For a strictly convex function $h : \mathcal{X} \to \mathbb{R}$ on a given space $\mathcal{X}$ (called the primal space) with (thus strictly monotonous) gradient $\nabla h : \mathcal{X} \to \mathcal{Z}$ ($\mathcal{Z}$ is called the dual space), the Bregman divergence (Bregman, 1967) $D_h : \mathcal{X} \times \mathcal{X} \to [0, \infty)$ from $\boldsymbol{x}$ to $\boldsymbol{x}'$ with respect to $h$ is defined as

$$D_h(\boldsymbol{x}' \parallel \boldsymbol{x}) = h(\boldsymbol{x}') - h(\boldsymbol{x}) - \langle \nabla h(\boldsymbol{x}), \boldsymbol{x}' - \boldsymbol{x} \rangle \tag{2}$$

The Bregman divergence is similar to a metric or distance (nonnegative, zero iff $\boldsymbol{x} = \boldsymbol{x}'$), except that in general it is not symmetric in its arguments and only satisfies a weaker version of the triangle inequality (whose exact form is not relevant here). $D_h$ is convex in its first argument but not necessarily in the second (Edelsbrunner & Wagner, 2017). The projection of an $\boldsymbol{x} \in \mathcal{X}$ on a closed and convex set $\mathbb{K} \subseteq \mathcal{X}$ w.r.t. to $D_h$ exists and is unique:

$$\Pi_{\mathbb{K}}(\boldsymbol{x}) = \min_{\boldsymbol{x}' \in \mathbb{K}} D_h(\boldsymbol{x}' \parallel \boldsymbol{x}). \tag{3}$$

The generic concepts are shown in the first column in Table 1; the other columns are examples. The squared Euclidean distance is a Bregman divergence if $h$ is chosen as the squared $L^2$ norm. More in-

---

[1]All threat models add another condition to ensure that the adversarial example $\boldsymbol{x}'$ does not exceed its natural range of pixels.

Table 1: Notation and context of our approach. The first column shows the generic concepts associated with the Bregman divergence and mirror descent. The second and third columns are known instantiations. The last column is our contribution and basis for a novel approach to robustness.

| Generic | Euclidean norm | KL divergence | Ours |
|---|---|---|---|
| Some space $\mathcal{X}$ | Euclidean space | Discrete distributions | Images |
| Base function $h : \mathcal{X} \to \mathbb{R}$ 
 *(strictly convex)* | $h(\boldsymbol{x}) = \frac{1}{2}\|\boldsymbol{x}\|_2^2$ | $h(\boldsymbol{p}) = \sum_i \boldsymbol{p}_i \log(\boldsymbol{p}_i)$ 
 *(Shannon entropy)* | $h = $ learned $\phi$ 
 (an ICNN) |
| Mirror map $\nabla h : \mathcal{X} \to \mathcal{Z}$ 
 *(strictly monotone)* | $\nabla h(\boldsymbol{x}) = \boldsymbol{x}$ | $\nabla h(\boldsymbol{p})_i = \log(\boldsymbol{p}_i)$ | $\Psi \approx \nabla h$ 
 (approximate gradient) |
| Inverse map $(\nabla h)^{-1} : \mathcal{Z} \to \mathcal{X}$ | $(\nabla h)^{-1}(\boldsymbol{z}) = \boldsymbol{z}$ | $(\nabla h)^{-1}(\boldsymbol{z})_i = e^{\boldsymbol{z}_i}$ | Fenchel conjugate $\overline{\Psi}$ |
| **Bregman Divergence** 
 $D_h(\boldsymbol{x}' \parallel \boldsymbol{x})$ | $\frac{1}{2}\|\boldsymbol{x}' - \boldsymbol{x}\|_2^2$ | $\sum_i \boldsymbol{q}_i \log \frac{\boldsymbol{q}_i}{\boldsymbol{p}_i}$ | $D_\phi$ 
 (learned divergence) |
| **Mirror descent** | PGD | Hedge algorithm | Ours |
| $\boldsymbol{z}^t = \nabla h(\boldsymbol{x}^t)$ 
 $\boldsymbol{z}^{t+1} = \boldsymbol{z}^t - \eta \nabla f(\boldsymbol{x}^t)$ 
 $\boldsymbol{x}^* = (\nabla h)^{-1}(\boldsymbol{z}^{t+1})$ 
 $\boldsymbol{x}^{t+1} = \Pi_{\mathbb{K}}(\boldsymbol{x}^*)$ | $\boldsymbol{x}^* = \boldsymbol{x}^t - \eta \nabla f(\boldsymbol{x}^t)$ 
 $\boldsymbol{x}^{t+1} = \Pi_{\mathbb{B}}(\boldsymbol{x}^*)$ | $\boldsymbol{p}_i^* = \boldsymbol{p}_i^t e^{-\eta \boldsymbol{l}_i}$ 
 $\boldsymbol{p}^{t+1} = \Pi_\Delta(\boldsymbol{p}^*)$ | $\boldsymbol{z}^t = \Psi(\boldsymbol{x}^t)$ 
 $\boldsymbol{z}^{t+1} = \boldsymbol{z}^t + \eta \nabla l(\boldsymbol{x}^t)$ 
 $\boldsymbol{x}^* = \overline{\Psi}(\boldsymbol{z}^{t+1})$ 
 $\boldsymbol{x}^{t+1} = \Pi_{\mathbb{S}}(\boldsymbol{x}^*)$ |

terestingly, if $h$ is the negative Shannon entropy, the associated Bregman divergence is the Kullback-Leibler (KL) divergence. Other examples of Bregman divergence include the Itakura–Saito distance, LeCam divergence, Brug divergence, Jeffreys Divergence, or Stein Divergence (Bauschke & Borwein, 1997; Stummer & Vajda, 2009; Harandi et al., 2014). The Bregman divergence is used if no suitable choice of a metric is available.

$L^p$ balls generalize to the Bregman divergence: the *Bregman ball* centered on $\boldsymbol{x}$ with radius $\epsilon$ is given by

$$\mathbb{B}_h(\boldsymbol{x}, \epsilon) = \left\{ \boldsymbol{x}' \in \mathcal{X} \mid D_h(\boldsymbol{x}' \parallel \boldsymbol{x}) \le \epsilon \right\}. \tag{4}$$

$\mathbb{B}_h$ is bounded and compact if $\mathcal{X}$ is closed but not necessarily convex (Edelsbrunner & Wagner, 2017).

**Mirror descent.** Mirror descent (Nemirovskij & Yudin, 1983) is a framework for optimizing functions $f : \mathcal{X} \to \mathbb{R}$ possibly constrained to a feasible convex set $\mathbb{K}$, $\min_{\boldsymbol{x} \in \mathbb{K}} f(\boldsymbol{x})$ given a suitable base function $h$ that defines a Bregman divergence. Mirror descent requires the gradient $\nabla h$ (called the *mirror map*) and the existence of $(\nabla h)^{-1}$ (called the *the inverse map*). The algorithm is iterative as shown in the first column in Table 1. After initializing $\boldsymbol{x}^0$ at any point in $\mathbb{K}$, each iteration $t$ consists of four steps: *(i)* mapping the current point $\boldsymbol{x}^t$ to a point in the dual space $\boldsymbol{z}^t = \nabla h(\boldsymbol{x}^t)$ through the mirror map, *(ii)* taking a gradient step of size $\eta$: $\boldsymbol{z}^{t+1} = \boldsymbol{z}^t - \eta \nabla f(\boldsymbol{x}^t)$, *(iii)* mapping $\boldsymbol{z}^{t+1}$ back to the primal space using the inverse map: $\boldsymbol{x}^* = (\nabla h)^{-1}(\boldsymbol{z}^{t+1})$, *(iv)* projecting $\boldsymbol{x}^*$ into the feasible set $\mathbb{K}$ w.r.t. $D_h$: $\boldsymbol{x}^{t+1} = \Pi_{\mathbb{K}}(\boldsymbol{x}^*)$ with (3).

As shown in Table 1, for the Euclidean divergence, mirror descent is exactly PGD. For the KL divergence it becomes the so-called hedge algorithm (Freund & Schapire, 1997). In this paper, as sketched in the fourth column, we will learn base functions $h$ that we call $\phi$ and associated divergences for common image corruptions and use them for AT as explained next.

## 3 GENERATING A BREGMAN DIVERGENCE FOR IMAGES

Motivated by the KL divergence in information theory we exploit the theory of Bregman divergence to derive new tools for robustness that match the geometry of image corruptions. Our high-level approach is outlined in the fourth column of Table 1. For a given type of corruption, we learn a base function $h = \phi$ that satisfies the properties to make $D_\phi$ a divergence. Mathematically, this $\phi$ will

play the same role as the Shannon entropy for KL divergence. We then instantiate mirror descent to solve the inner maximization problem in (1) and thus enable AT with $D_\phi$. Formally, the challenge is to learn a $\phi$ with the following properties:

(i) $\phi$ is convex and differentiable, and thus $D_\phi$ a divergence;

(ii) $\nabla\phi(\boldsymbol{x})$ and $(\nabla\phi)^{-1}(\boldsymbol{x})$ have to be (approximately) computable to execute mirror descent.

Hand-engineering an eligible and performant $\phi$ (e.g., by stacking feature extractors) is likely to be a daunting task. We propose to model $\phi$ as a deep neural network with a particular architecture: the *input convex neural network (ICNN)* (Amos et al., 2017) for which we design a self-supervised learning algorithm. The details are explained next.

## 3.1 CONVEX ARCHITECTURE

Based on the work by Amos et al. (2017), we define $\phi$ as an ICNN. The architecture is an $L$-layered deep neural network with activations $\boldsymbol{z}^l$ given by:

$$
\begin{aligned}
\boldsymbol{z}^1 &= g^0(\boldsymbol{W}^0\boldsymbol{x} + \boldsymbol{b}^0), \\
\boldsymbol{z}^l &= g^{l-1}(\boldsymbol{W}^{l-1}\boldsymbol{x} + \boldsymbol{V}^{l-1}\boldsymbol{z}^{l-1} + \boldsymbol{b}^{l-1}) \ \text{ for } l = 2,\dots,L.
\end{aligned}
\tag{5}
$$

The output is $\phi(\boldsymbol{x}) = \boldsymbol{z}^L$. All the weights $\boldsymbol{W}^l$ and $\boldsymbol{V}^l$ and the biases $\boldsymbol{b}^l$ are learnable parameters. The function $\phi$ is convex provided that all $\boldsymbol{V}^l$ are non-negative and all the activation functions $g^l$ are convex and non-decreasing (Amos et al., 2017, Proposition 1). We set all the activation functions $g^l$ to be the continuously differentiable exponential linear unit (CELU) (Barron, 2017) and the linear layers as convolutions. Once we have $\phi$, we numerically approximate the evaluation of the mirror map $\Psi(\boldsymbol{x}) \approx \nabla\phi(\boldsymbol{x})$ using automatic differentiation (Paszke et al., 2017).

## 3.2 INVERSE MAP

Since $\Psi$ is a gradient of a neural network, its inverse $\Psi^{-1}$ is not readily available. Our solution leverages the Fenchel conjugate (Fenchel, 1949) $\overline{\phi} : \mathcal{Z} \to \mathbb{R}$ of $\phi$, which exists for convex $\phi$, is again convex, and defined as:

$$
\overline{\phi}(\boldsymbol{z}) = \max_{\boldsymbol{x}} \ \langle \boldsymbol{x}, \boldsymbol{z} \rangle - \phi(\boldsymbol{x}).
\tag{6}
$$

If $\phi$ is of so-called *Legendre type* (i.e., proper closed, essentially smooth and essentially strictly convex (Rockafellar, 1970)), then Fenchel (1949) states that $(\nabla\phi)^{-1} = \nabla\overline{\phi}$. Inspired by this equation, we define the conjugate $\overline{\phi}$ again as an ICNN with the exact same architecture as $\phi$ in (5). Given $\Psi$, it is trained by minimizing:[2]

$$
\min_{\overline{\phi}, \overline{\Psi}} \mathbb{E}_{\boldsymbol{x}\sim\mathcal{D}} \left[ ||\overline{\Psi}(\Psi(\boldsymbol{x})) - \boldsymbol{x}||_2 \right].
\tag{7}
$$

Now $\overline{\Psi}(\boldsymbol{x}) \approx \nabla\overline{\phi}(\boldsymbol{x})$ is again computed using automatic differentiation and approximates $(\nabla\phi)^{-1}(\boldsymbol{x})$ as desired.

## 3.3 TRAINING ALGORITHM

A real-world corruption $\tau(\boldsymbol{x})$ (like blurred or with changed contrast) typically lies at a large $L^2$ distance $\epsilon$ (say 10) of the clean image $\boldsymbol{x}$ and thus an $L^2$-based attack with this $\epsilon$ would not find it but instead an extremely noisy one $\tilde{\boldsymbol{x}}$ at similar distance which would likely not be recognizable by a human (Fig. 1, left). Also, AT does not converge for large $\epsilon$ and typically very small $\epsilon$ around $0.1$ are used (Hendrycks & Dietterich, 2019; Ford et al., 2019; Xie et al., 2020; Kang et al., 2019; Kireev et al., 2022).

Our basic idea is to train $\phi$ such that, with respect to the induced Bregman divergence, a suitable Bregman ball $\mathbb{B}_\phi$ includes $\tau(\boldsymbol{x})$ while excluding noisy images at much closer $L^2$ distance (Fig. 1, right). To do so, we first need a way to sample random images at a given mean distance from a clean image:

---

[2]In this expression, $\overline{\Psi}$ is not an explicit neural network but rather a gradient of a neural network ($\overline{\phi}$) computed w.r.t. the input.

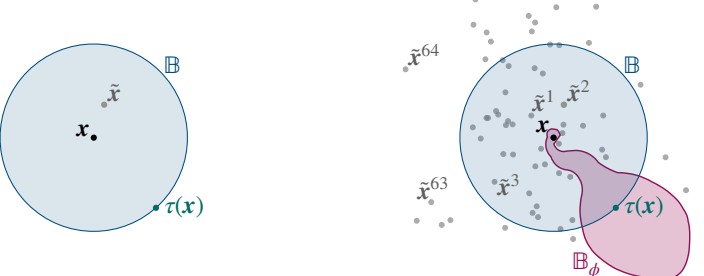

Figure 1: A two-dimensional cartoon visualizing our approach. Left: a clean image $x$ corrupted to $\tau(x)$ whose inclusion in an $L^2$ ball $\mathbb{B}$ requires a large radius. An attack with this distance would yield an extremely noisy $\tilde{x}$. Right: our learned Bregman distance yields balls that can include $\tau(x)$ while excluding noisy images $\tilde{x}$ (here 64 many) at much closer $L^2$ distance.

**Lemma 1** *When sampling the isotropic Gaussian random variable*

$$\tilde{\mathrm{x}} = x + d\,\frac{\Gamma\left(\frac{n}{2}\right)}{\sqrt{2}\,\Gamma\left(\frac{n+1}{2}\right)}\,||\tau(x) - x||_2\,\delta\,,\;\;\delta \sim \mathcal{N}(0, I_n), \tag{8}$$

*where $x$ is a fixed clean image, $d \in (0,1)$ is a hyperparameter and $\Gamma$ is the gamma function, we have*

$$\mathbb{E}\left[||\tilde{\mathrm{x}} - x||_2\right] = d||\tau(x) - x||_2. \tag{9}$$

The proof is given in Appendix A.1.

Fig. 1 (right) shows $m = 64$ such samples $\{\tilde{x}^i\}_{i=1}^m$ using $d = 1$. Next, we force each of their divergences $D_\phi(\tilde{x}^i \parallel x)$ to be larger than $D_\phi(\tau(x) \parallel x)$ or equivalently $-D_\phi(\tau(x) \parallel x) > -D_\phi(\tilde{x}^i \parallel x)$. We propose the following *Bregman loss* $l_B(x; \phi, \Psi)$ to enforce all these $m$ inequalities at once:

$$l_B(x; \phi, \Psi) = -\log\frac{e^{-D_\phi(\tau(x)\|x)}}{e^{-D_\phi(\tau(x)\|x)} + \sum_i e^{-D_\phi(\tilde{x}^i\|x)}}. \tag{10}$$

$l_B(x; \phi, \Psi)$ can be interpreted as a cross entropy where the logits vector is the negative of Bregman divergences $\left[-D_\phi(\tau(x) \parallel x), -D_\phi(\tilde{x}^1 \parallel x), ..., -D_\phi(\tilde{x}^m \parallel x)\right]$ and the ground truth class always corresponds the first entry. Then, we learn $\phi$ by minimizing:

$$\min_{\phi,\Psi}\mathbb{E}_{x\sim\mathcal{D}}\left[l_B(x; \phi, \Psi)\right]. \tag{11}$$

After a successful training where $D_\phi(\tau(x) \parallel x) < D_\phi(\tilde{x}^i \parallel x)$ for all $i = 1, ..., m$, the Bregman ball $\mathbb{B}_\phi(x, D_\phi(\tau(x) \parallel x))$ contains the transformed image $\tau(x)$ by definition but does not contain any of the noisy images $\{\tilde{x}^i\}_{i=1}^m$ (Fig. 1, right).

### 3.4 BREGMAN-BASED SEMANTIC ATTACK

Given a learned Bregman divergence, we define the neighborhood of a clean image $x$ as the intersection of a Bregman ball and fixed $L^2$ ball:

$$\mathbb{S}(x) = \mathbb{B}_\phi(x, \epsilon) \cap \mathbb{B}(x, \epsilon_{max}). \tag{12}$$

Here, as expected, $\epsilon$ is a parameter used in the adversarial attacks; $\epsilon_{max}$ is an empirical large value, typically chosen a hundred times bigger than the usual box radii used for the $L^2$-based ATs. We empirically found this $L^2$ bound to be necessary. The reason is that most random samples used for training with (11) are near $x$ w.r.t $L^2$ and thus $D_\phi$ might assign also small values in "under-explored" regions that are far from $x$. As a result, a Bregman ball of radius $D_\phi(\tau_s(x) \parallel x)$ may include these regions. Experimentally, we found that imposing a large $L^2$-bound on $\mathbb{B}_\phi$ eliminates this problem.

---

**Algorithm 1** Bregman-based semantic attack

---

1: $\boldsymbol{x}' \leftarrow \boldsymbol{x}$ ▷ Initialization
2: **for** $t = 1, ..., T$ **do**
3:     $\eta \leftarrow \epsilon 10^{-4t/T}$
4:     $\boldsymbol{x}' \leftarrow \overline{\Psi}\left(\Psi(\boldsymbol{x}') + \eta\nabla_{\boldsymbol{x}}l(\boldsymbol{x}', y; \theta)\right)$ ▷ The mirror descent update explained in Sec. 2
5:     $\boldsymbol{x}' \leftarrow \boldsymbol{x} - \epsilon_{max}(\boldsymbol{x} - \boldsymbol{x}')/\|\boldsymbol{x}' - \boldsymbol{x}\|_2$ ▷ Projecting $\boldsymbol{x}'$ into $\mathbb{B}(\boldsymbol{x}, \epsilon_{max})$
6:     $a, b \leftarrow 0, 1$
7:     **while** $D_\phi(\boldsymbol{x}' \| \boldsymbol{x}) > \epsilon$ **do** ▷ Projecting $\boldsymbol{x}'$ into $\mathbb{B}_\phi(\boldsymbol{x}, \epsilon)$
8:         $m \leftarrow (a + b)/2$
9:         $\boldsymbol{x}' \leftarrow \boldsymbol{x} + m(\boldsymbol{x}' - \boldsymbol{x})$
10:         $a \leftarrow m$ **if** $D_\phi(\boldsymbol{x}' \| \boldsymbol{x}) > \epsilon$ **else** $b \leftarrow m$
11:     $\boldsymbol{x}' \leftarrow \texttt{clip}(\boldsymbol{x}', 0, 1)$ ▷ Projecting $\boldsymbol{x}'$ into $[0, 1]^n$
12: **return** $\boldsymbol{x}'$ ▷ $\boldsymbol{x}'$ is potentially misclassified

---

Our *Bregman-based semantic attack* follows the mirror descent as outlined in the fourth column of Table 1 and detailed in Algorithm 1. The projection of $\boldsymbol{x}'$ into $\mathbb{S}$ is done as a projection into $\mathbb{B}(\boldsymbol{x}, \epsilon_{max})$ followed by a projection into $\mathbb{B}_\phi(\boldsymbol{x}, \epsilon)$. Since the latter has no closed-from expression, we approximate it by a binary search over the segment having $\boldsymbol{x}$ and $\boldsymbol{x}'$ as endpoints (lines 7–11 in Algorithm 1).

## 4 RELATED WORK

**Corruption robustness via data augmentation.** Much of the prior literature on corruption robustness aims to improve out-of-distribution generalization by using simulated and augmented images for training. Many such data augmentation techniques are based on creating synthetic training examples through mixing pairs of training images and their labels. This is achieved for example by linear weighted blending of images (Zhang et al., 2018a) or by cutting and pasting parts of an image onto another (Yun et al., 2019). Researchers also fused images based on masks computed through frequency spectrum analysis (Harris et al., 2021), based on adaptive masks (Liu et al., 2022) or based on model-generated features (Walawalkar et al., 2020). Other works considered a hybrid version of these mixing methods (Park et al., 2022), a stochastic version of them (Park et al., 2022), an ensemble of them (Yin et al., 2022) or a concurrent combination of them (Liang et al., 2023).

**Adversarial attacks without $L^p$ norms.** Another line of related work focuses on adversarial image perturbations that are not constrained by $L^p$ norms. Hsiung et al. (2023) introduces semantic adversarial attacks that target image transformation parameters instead of image pixels. Similarly, Engstrom et al. (2019) targets spatial transformations. Hosseini & Poovendran (2018) manipulates the hue and saturation components in the hue saturation value (HSV) color space to create adversarial examples. In addition to colorization, Bhattad et al. (2019) also tweaked texture of objects within images. (Shamsabadi et al., 2020) modified colors within the invisible range. Some works altered the semantic features of images through conditional generative models (Joshi et al., 2019) or conditional image editing(Qiu et al., 2020).

**Robustness via learned similarity metric.** The closest related work adopts the so-called learned perceptual image patch similarity (LPIPS) to study robustness. LPIPS is a weighted sum of the $L^2$ of the feature maps taken from the activation layers of a trained convolutional network:

$$\text{LPIPS}(\boldsymbol{x}, \boldsymbol{x}') = \sum_l w_l \|\omega_l(\boldsymbol{x}) - \omega_l(\boldsymbol{x}')\|_2 \tag{13}$$

where $\omega_l$ is the feature map up to the $l$-layer and $w_l$ weighs the contribution of the layer $l$. Wang et al. (2021) and Luo et al. (2022) propose an attack similar to (Carlini & Wagner, 2017) by adding the LPIPS along with the $L^p$ norm. Differently, Kireev et al. (2022) and Laidlaw et al. (2021) used LPIPS as a function to define the set of similar images (refer to Sec. 2 for notation context): $\mathbb{S}(\boldsymbol{x}) = \{\boldsymbol{x}' \in \mathbb{R}^n | \text{LPIPS}(\boldsymbol{x}, \boldsymbol{x}') \leq \epsilon\}$. Since the projection into this LPIPS-based set does not admit a closed-from expression, solving the inner maximization problem of (1) (i.e., performing the adversarial attack) requires approximation (Laidlaw et al., 2021) or relaxation (Kireev et al., 2022).

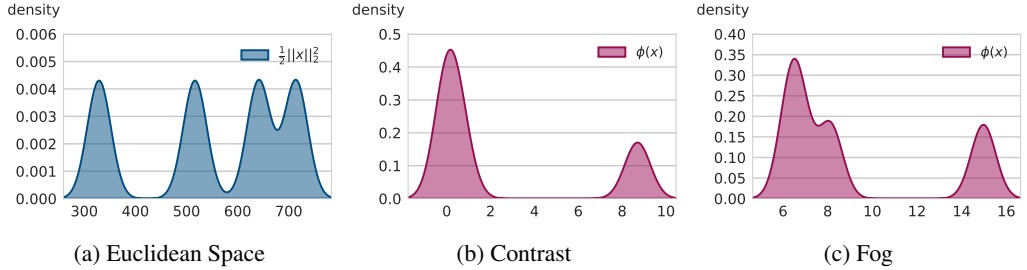

(a) Euclidean Space          (b) Contrast          (c) Fog

Figure 2: Distribution of Bregman divergence's base functions over 10,000 CIFAR-10 test set images. Two trained base functions $\phi$, (b) for contrast and (c) for fog, are compared against (a) their counterpart in the PGD setting, the half of norm $L^2$ squared (see Table 1).

The resulting attacks and their associated AT have been proven effective to train robust models against common image corruptions. We compare against LPIPS in our experiments.

## 5  EVALUATION

We perform experiments on CIFAR-10 (Krizhevsky et al., 2009) and the common corruption dataset CIFAR-10-C (Hendrycks & Dietterich, 2019). For the classification model, we use the PreAct ResNet-18 architecture (He et al., 2016) the same used by Kireev et al. (2022). The image corruptions are from (Hendrycks & Dietterich, 2019) and come with severities from 1 to 5. Our focus is on the corruptions of contrast and fog which have been found the most challenging in (Ford et al., 2019; Kireev et al., 2022).

**Training the Bregman divergence.** Both $\phi$ and $\overline{\phi}$ have the same architecture, an ICNN with 12 convolutional layers followed by 4 fully connected layers. The mirror map and the inverse mirror are numerically approximated using `autograd.grad` from PyTorch's automatic differentiation engine (Paszke et al., 2017). As an initialization phase, we first train $\phi$ and $\overline{\phi}$ such that $\Psi$ and $\overline{\Psi}$ approximate the identity function (so initially $\overline{\Psi} = \Psi^{-1}$ holds) on uniformly drawn samples from the usual range of pixels $[0, 1]^n$:

$$\min_{\phi, \Psi} \mathbb{E}_{\boldsymbol{x} \sim \mathcal{U}([0,1]^n)} \left[ ||\Psi(\boldsymbol{x}) - \boldsymbol{x}||_2 \right], \quad \min_{\overline{\phi}, \overline{\Psi}} \mathbb{E}_{\boldsymbol{x} \sim \mathcal{U}([0,1]^n)} \left[ ||\overline{\Psi}(\boldsymbol{x}) - \boldsymbol{x}||_2 \right]. \tag{14}$$

This training is performed for 10,000 steps using the Adam optimizer (Kingma & Ba, 2014) with a learning rate of 0.001 and a weight decay of $10^{-10}$.

Next, for a given corruption $\tau$, we train $\phi$ with (11) while randomly sampling its severity for each image at each epoch. The hyperparameter $d$ for sampling noisy images defined in (8) is uniformly sampled in $[10^{-7}, 10^{-1}]$. Since the fraction of $\Gamma$ functions in (8) is difficult to compute, we replace it empirically by $1/\sqrt{n}$. Doing so yields $\sqrt{\mathbb{E}\left[||\tilde{\mathbf{x}} - \boldsymbol{x}||_2^2\right]} = d||\tau_s(\boldsymbol{x}) - \boldsymbol{x}||_2$ instead of Lemma 1 (Appendix A.1). We use $m = 31$ samples per clean image while a batch consists of 16 clean images.

The training is performed for 20 epochs using the AdamW optimizer (Loshchilov & Hutter, 2017) with a learning rate of 0.0001 and a weight decay of $10^{-11}$. After each update of $\phi$ (according to Equ. 11), we also update $\overline{\phi}$ (according to Equ. 7). Finally, we freeze the parameters of $\phi$ and continue training $\overline{\phi}$ for additional 20 epochs.

**Results of the Bregman divergence.** We first inspect the learned base functions $\phi$, i.e., our trained "entropy" for an image corruption. The distribution of its outputs on the test set is shown in Fig. 2. We notice that the trained based functions have modalities with different shapes and heights compared to the $L^2$ norm, the base function in the $L^2$-based threat models (see Table 1).

Next we show in Fig. 3 that the learned divergence $D_\phi$ agrees with Fig. 1. To do so we consider, for the entire test set of 10,000 clean images, noisy images (blue, one per clean image) and the set of contrast-corrupted images (red). We compute the distribution of their $L^2$ distances to the clean image in Fig. 3a. It is small for the noisy images (by construction) and large for the corrupted

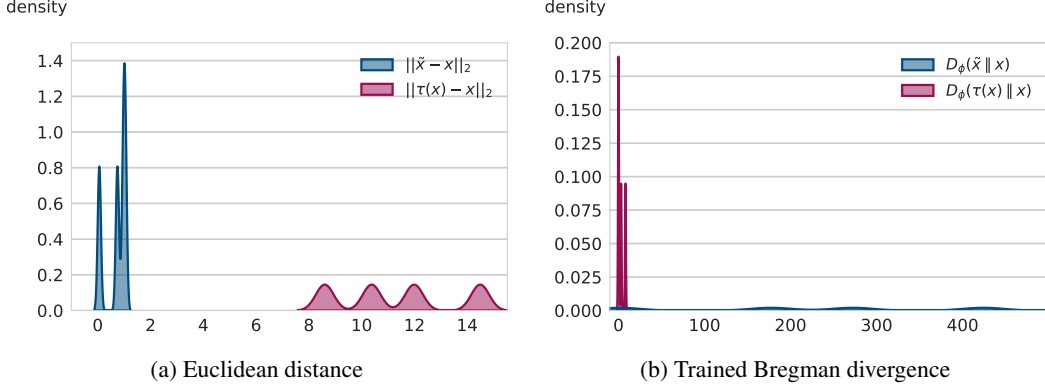

(a) Euclidean distance  (b) Trained Bregman divergence

Figure 3: Distribution of (a) the Euclidean distance and (b) our trained Bregman divergence for noisy images $\tilde{x}$ (blue) and contrast-corrupted images $\tau(x)$ (red) over 10,000 CIFAR-10 test set images.

ones. Fig. 3b shows the distribution of their learned divergences to the clean image.[3] Here, the corrupted images are considered close but the noisy ones far, which shows that the learned Bregman divergence is semantically meaningful and works as expected.

**Adversarial training.** After training a Bregman divergence to be semantically meaningful for a corruption $\tau$ and using it in the definition of the threat model, we can run an AT using our Bregman-based semantic attack. We call this procedure the *Bregman-based adversarial training (BAT)*. For a corruption type $\tau$ we write BAT($\tau$).

We compare against the relaxed LPIPS AT (RLAT) (Kireev et al., 2022) without any further fine-tuning of the training parameters. For a fair comparison, we set the number of iteration of our attack to $T = 1$ to match the one-step attack used in RLAT. Fig. 4 shows a sample of the resulting images for contrast corruptions. The training is performed using the SGD optimizer for 150 epochs with a learning rate of 0.1 that decays by a factor of 10 each 50 epochs, a batch size of 128, and a weight decay of 0.0005. These are the same hyperparameters for which RLAT performs the best. The RLAT radius is taken to be 0.08. We also compare against the $L^2$ PGD AT with the radius of 0.1, which Kireev et al. (2022) found the most effective for corruption robustness. We emphasize that there is no meaning in the relation of the magnitudes between $L^2$ radius values and Bregman ball radius values.

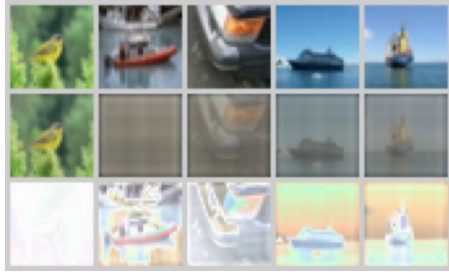

Figure 4: Samples from training images (first row), adversarial examples for contrast corruption found by our Bregman-based attack (second row) and the pixel-wise difference.

We run BAT for different values of Bregman ball radius $\epsilon$ from 0.025 to 20 while fixing $\epsilon_{max} = 10$ in (12). We notice that the accuracy of the resulting classifier is not very sensitive to the choice of $\epsilon$ unlike in $L^p$- and LPIPS-based threat models (Kireev et al., 2022).

For contrast corruptions, the best accuracy is achieved with $\epsilon = 0.2$ which we use in the comparison in Table 2. The standard accuracy is on the clean data set. The columns show the corruption accuracy for different severities and the last is their average. BAT preserves the standard accuracy (a slight drop of 0.2%) while maintains high accuracies even under high corruption severities where the other methods start to fail.

Additionally, we trained for fog and zoom blur corruptions and report the average results in Table 3 together with the prior contrast corruption. Fog is again best by a margin, whereas for zoom blur we perform about 5% worse. Surprisingly our training for contrast also performs best on fog and well

---

[3]The mean, std, min and max of the distributions are reported in Appendix B.

Table 2: Comparison of corruption robustness of models trained under different regimes.

| | Standard accuracy | Contrast | | | | | Average |
|---|---|---|---|---|---|---|---|
| | | $s = 1$ | $s = 2$ | $s = 3$ | $s = 4$ | $s = 5$ | |
| Standard training | 94.82 | 94.25 | 90.57 | 86.35 | 76.42 | 34.72 | 76.46 |
| $L^2$-based AT | 93.52 | 91.68 | 82.96 | 72.31 | 51.43 | 21.26 | 63.92 |
| RLAT | 93.27 | 91.47 | 82.32 | 70.65 | 48.35 | 21.58 | 62.87 |
| BAT(contrast) | 94.62 | 94.38 | 93.74 | 93.07 | 91.58 | 84.04 | 91.36 |

Table 3: Corruption robustness of the standard-trained model against adversarially trained models under $L^2$, RLAT, and BAT for different corruptions.

| | Standard | Contrast | Fog | Zoom Blur |
|---|---|---|---|---|
| Standard training | 94.82 | 76.46 | 87.04 | 77.22 |
| $L^2$-based AT | 93.52 | 63.92 | 77.47 | 85.87 |
| RLAT | 93.27 | 62.87 | 77.00 | 85.88 |
| BAT(contrast) | 94.62 | 91.36 | 91.47 | 80.11 |
| BAT(fog) | 94.66 | 79.53 | 89.20 | 80.42 |
| BAT(zoom blur) | 93.86 | 89.60 | 90.13 | 80.62 |

for zoom blur. One hint is that the three corruptions are similar in nature but this behavior requires further investigation.

**Limitation and discussion.** The current main limitation of our approach is that it is specific to the corruption type, and, as shown by zoom blur, does not always perform best. Further, there is an overhead in first training for a valid divergence before starting the AT. On the other hand we could significantly improve robustness on contrast and fog where the other methods fail to improve over standard training. Also, our method produces adversarial examples unlike the work by Kireev et al. (2022) that perturbs the feature space with no mechanism to produce an associated image. Finally, we see value in the theoretical underpinning, which yields desirable properties (e.g., the Bregman ball is compact unlike the LPIPS-based sets) and the well-established mirror descent. In fact, the learned base function and divergence may be interesting for other applications. We hope that scaling to larger convex architectures (the ones presented in this work are tiny; contain only 2M parameters) would yield base functions $\phi$ that generalize to large sets of corruption types.

## 6    CONCLUSION

We presented a novel approach and tool set to improve corruption robustness of neural networks for image classification. The key idea was to first learn a similarity measure that semantically captures a considered corruption and that is also theoretically sound by instantiating a Bregman divergence. Doing so gave access to executing mirror descent and thus adversarial attacks and training. The main, and significant, challenge in our work was to ensure that the learned base functions underlying the divergence satisfy the properties required by the theory.

Our results are prototypical and can only be considered a first step but we consider them in strong support of our novel contribution. Specifically, we demonstrated that the learned divergence measures similarity as intended and we could demonstrate a significantly improved corruption robustness for two corruptions on which prior work failed. Thus we see great potential when using large network models and training sets. Finally, due to the underlying theory, our work is not specific to images and learned divergences may find applications outside the scope of this paper.

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

# A    PROOFS

## A.1    LEMMA 1

Let $\boldsymbol{x} \in \mathbb{R}^n$ a fixed image and $\tilde{\text{x}}$ a random variable defined as follows with $\mu > 0$:

$$\tilde{\text{x}} = \boldsymbol{x} + \mu\boldsymbol{\delta} \ , \ \ \boldsymbol{\delta} \sim \mathcal{N}(0, \boldsymbol{I}_n), \tag{15}$$

The random variable $\tilde{\text{x}}$ is a gaussian because it is a linear combination of gaussians ($\boldsymbol{x}$ is fixed).

$$
\begin{aligned}
||\tilde{\text{x}} - \boldsymbol{x}||_2 &= \sqrt{\sum_i (\tilde{\text{x}}_i - \boldsymbol{x}_i)^2} \\
&= \mu\sqrt{\sum_i \boldsymbol{\delta}_i^2} \\
&= \mu\sqrt{\text{z}}
\end{aligned}
\tag{16}
$$

z is a chi square distribution of degree $n$ with density:

$$p_{\text{z}}(z) = \frac{z^{n/2-1}e^{-z/2}}{2^{n/2}\Gamma(\frac{n}{2})} \tag{17}$$

We defined the variable $\text{u} = f(\text{z}) = \sqrt{\text{z}}$, $\text{u} \geq 0$. The density of u can be computed by the change or variable formula:

$$
\begin{aligned}
p_{\text{u}}(u) &= p_{\text{z}}(f^{-1}(u))\left|\frac{dz}{du}\right| \\
&= \frac{u^{n-1}e^{-u^2/2}}{2^{n/2-1}\Gamma(\frac{n}{2})}
\end{aligned}
\tag{18}
$$

Next, we compute the expectation of u:

$$
\begin{aligned}
\mathbb{E}(\text{u}) &= \int_0^\infty u p_{\text{u}}(u)du \\
&= \frac{1}{2^{n/2-1}\Gamma(\frac{n}{2})}\int_0^\infty u^n e^{-u^2/2}du \\
&= \frac{\sqrt{2}}{\Gamma(\frac{n}{2})}\int_0^\infty t^{(n-1)/2}e^{-t}dt \qquad \text{(by substituting} \ \ u = \sqrt{2t}) \\
&= \frac{\sqrt{2}}{\Gamma(\frac{n}{2})}\Gamma(\frac{n+1}{2}) \qquad\qquad\qquad \text{(by definition of } \Gamma)
\end{aligned}
\tag{19}
$$

So we have:

$$\mathbb{E}(||\tilde{\text{x}} - \boldsymbol{x}||_2) = \mathbb{E}(\mu\text{u}) = \mu\frac{\sqrt{2}\Gamma(\frac{n+1}{2})}{\Gamma(\frac{n}{2})} \tag{20}$$

In Equ. 15, we set:

$$\mu = d \ \frac{\Gamma\left(\frac{n}{2}\right)}{\sqrt{2}\ \Gamma\left(\frac{n+1}{2}\right)} \ ||\tau(\boldsymbol{x}) - \boldsymbol{x}||_2 \tag{21}$$

to obtain the same expression as in Lemma 1

## A.2 AN ALTERNATIVE TO LEMMA 1

For large values of $n$, the constants $\Gamma\left(\frac{n}{2}\right)$ and $\Gamma\left(\frac{n+1}{2}\right)$ are intractable. Here we propose an alternative. With the same notation as above :

$$\mathbb{E}(||\tilde{\mathrm{x}} - \boldsymbol{x}||_2^2) = \mu^2 \mathbb{E}(\mathrm{z}) = \mu^2 n. \tag{22}$$

In Equ. 15, we set:

$$\mu = d\,\frac{1}{\sqrt{n}}\,||\tau(\boldsymbol{x}) - \boldsymbol{x}||_2 \tag{23}$$

and obtain:

$$\sqrt{\mathbb{E}(||\tilde{\mathrm{x}} - \boldsymbol{x}||_2^2)} = d||\tau(\boldsymbol{x}) - \boldsymbol{x}||_2. \tag{24}$$

## B DETAILS ABOUT THE DIVERGENCE EVALUATION

In Sec. 5, we have shown that our trained Bregman divergence is semantically meaningful as it considers noisy image far off clean images and the corrupted images closer even when the $L^2$ says otherwise. The distribution of these Bregman and $L^2$ values that are shown in Fig. 3 are further described in Tab. 4.

Table 4: Description of the distributions from Fig. 3.

|      | $||\tau(\boldsymbol{x}) - \boldsymbol{x}||_2$ | $||\tilde{\boldsymbol{x}} - \boldsymbol{x}||_2$ | $D_\phi(\tau(\boldsymbol{x}) \,\|\, \boldsymbol{x})$ | $D_\phi(\tilde{\boldsymbol{x}} \,\|\, \boldsymbol{x})$ |
|------|------|------|------|------|
| mean | 11.36 | 0.70 | 2.79 | 218.69 |
| std  | 2.16 | 0.39 | 3.31 | 153.34 |
| min  | 8.60 | 0.06 | $10^{-6}$ | 1.11 |
| max  | 14.49 | 1.04 | 8.11 | 423.90 |

## C THE EVOLUTION OF THE TRAINING/VALIDATION LOSS

Figure 5 illustrates the convergence of the Bragman training under the settings of Sec. 5.

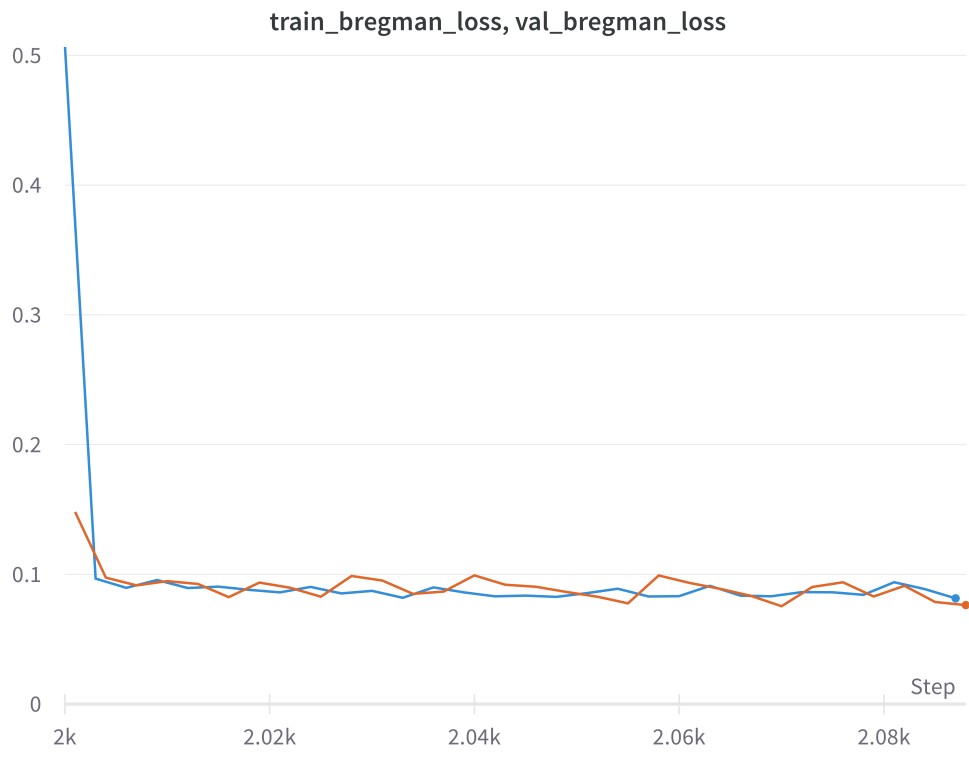

Figure 5: The evolution of the Bregman loss $l_B(\boldsymbol{x}; \phi, \Psi)$ for the training and the validation set across optimization steps.

