# OpenReview forum: "Robustness via learned Bregman divergence"
_ICLR.cc/2024/Conference — Submitted to ICLR 2024_

### Official Review · Reviewer_qLEt · 2023-10-29

**Soundness:** 2 fair
**Presentation:** 3 good
**Contribution:** 2 fair
**Rating:** 3
**Confidence:** 4

**Summary:**

The paper uses Bregman divergence for adversarial learning, which the authors argue to be more robust and can be more accurate.

**Strengths:**

The exposition is nice and the manuscript is easy to follow.

**Weaknesses:**

Adversarial learning is a fast-growing field. That being said, I failed to see the technical novelty of the proposed approach: the results appear to be straightforward and the technical contributions are limited. There is only one lemma and an algorithm in the current manuscript, making the paper more like in an engineering manner.

**Questions:**

See my comments in the limitations.

---

> ### Author Response · Authors · 2023-11-17
> **Inappropriate review**
>
> We reported this review as inappropriate.

---

> ### Comment · Area_Chair_zt33 · 2023-11-20
>
> Dear Reviewer qLEt and Authors,
>
> To Reviewer qLEt: your review does not follow guidelines for a concrete technical and factual evaluation of the work or provide actionable items for the authors. Please expand your review by providing more factual arguments. Otherwise, please reduce your confidence score.
>
> To Authors: currently, there is nothing to address for this review.
>
> Best,
> Your AC

---

### Official Review · Reviewer_brGx · 2023-10-30

**Soundness:** 2 fair
**Presentation:** 3 good
**Contribution:** 2 fair
**Rating:** 3
**Confidence:** 4

**Summary:**

The study examines the use of Bregman divergence as an alternative to the $L^p$ norm for measuring the distance between benign and corrupted samples. It also explores the application of mirror descent, based on Bregman divergence, to identify corruptions for adversarial training. This is accomplished by constructing two convex neural networks that approximate the gradient of the distance-generating function and its inverse. Training the networks involves utilizing a Bregman loss. Additionally, a semantic attack is proposed using Bregman divergence to generate corrupted examples, which are then employed in adversarial training to enhance corruption robustness.

**Strengths:**

1. The organization of this paper is clear and logical. The motivation and procedures of the proposed method are well-described.
2. The proposed method is novel, and the experimental results demonstrate its superior performance compared to the baseline methods across different corruption severities.

**Weaknesses:**

1. The claim that $(\nabla\phi)^{-1}=\nabla\bar{\phi}$ is based on the assumption that ICNN is of the Legendre type. However, this assumption has not been verified. The proposed numerical method for approximating the inverse $(\nabla\phi)^{-1}$ appears unrelated to Fenchel's (1949) result. According to Equation (7), the minimization problem $\min_f \Vert f(\nabla\phi(x)) - x\Vert_2$ should yield $f=(\nabla\phi)^{-1}$, but it is not demonstrated that the L2 distance is the optimal choice.
2. The evaluation in this paper has limitations. For instance, it remains unclear how the Bregman divergence training converges. Additionally, it is uncertain whether the method can scale to larger images, such as ImageNet. How does the choice of sampling numbers impact the approximation?
3. The Bregman-based attack finds the projection point $x'$ in the intersection of $D$, which is the intersection of $B_{\phi}$ and $B$. According to the algorithm, the projection is performed by projecting onto $B$ first, and then onto $B_{\phi}$. However, it seems unlikely that the resulting projected point will lie in the intersection.
4. Although Lemma 1 states that $d$ lies in the range $(0, 1)$, it is unclear why Figure 1 is generated when $d=1.
5. In the experiment, the fraction of $\Gamma$ functions is defined as $1/\sqrt{n}$. The purpose of introducing $\mu$ as the fraction of Gamma functions in Lemma 1 is not clear.
6. Based on my understanding, each iteration of mirror descent involves one forward pass and one backward pass for the input $x^t$. It is unclear how the computation cost of Bregman divergence training is determined.
7. The notations in the equations lack consistency, such as the font used for variables like $\tilde{x}$ and $u$.

**Questions:**

See weaknesses

---

> ### Author Response · Authors · 2023-11-16
>
> We have addressed your questions/concerns; we consider some as misunderstandings, other clarifications we will incorporate.
>
>
> $\def\vy{\boldsymbol{y}}$
> $\def\vx{\boldsymbol{x}}$
> $\def\vz{\boldsymbol{z}}$
> $\def\vu{\boldsymbol{u}}$
>
>
> We argue that our notation is consistent; indeed, we followed the conference recommendations provided in the ICLR template. For example, the font in `\bm{x}` $\vx$ indicates a vector where the font in `\textnormal{x}` $\textnormal{x}$ is for random variables. We also made these distinctions clear throughout the text.
>
> > "The claim that $(\nabla\phi)^{-1}=\nabla\overline\phi$ is based on the assumption that ICNN is of the Legendre type. However, this assumption has not been verified"
>
> We did not claim that ICNNs are of Legendre type (it would require proving that the limit along the boundary of the interior of the domain of definition of $\nabla\phi$ is $\infty$ which is challenging for an arbitrarily complex $\phi$). We did not claim that this equality holds for ICNNs. We only used this fact to motivate the approximation which we wrote as $\overline\Psi(\boldsymbol{x}) \approx \nabla\overline\phi(\boldsymbol{x})$.
>
> > "The proposed numerical method for approximating the inverse appears unrelated to Fenchel's (1949) result"
>
> The result of Fenchel (1949) is important for our work: it provides the insight that the inverse of $\Psi$ is the gradient of a another convex function (in particular the conjugate). We have relied on this to model $\overline\Psi$ as a gradient of an ICNN, which is convex by our construction. This inspiration is discussed right above Equation 7.
>
> > " ... but it is not demonstrated that the L2 distance is the optimal choice."
>
> Training for this objective (the mean square error) is a design choice that we made and it is a common choice, e.g., also widely used when training autoencoders. The choice gave good results as detailed in Sec.5. We did not claim its optimality and it can be replaced by alternatives if suggested.
>
> > "...it remains unclear how the Bregman divergence training converges."
>
> The training **converges** nicely without over-fitting. The evolution of the training and the validation loss across optimization steps is added in Appendix C.
>
> > "... the projection is performed by projecting onto $B$ first, and then onto $B_\phi$. However, it seems unlikely that the resulting projected point will lie in the intersection."
>
> $\def\epsm{{\epsilon_\text{max}}}$
>
> $\def\Bf{{\mathbb{B}_\phi(\vx, \epsilon)}}$
>
> $\def\B{{\mathbb{B}(\vx, \epsilon_\text{max})}}$
>
>
> The result of this projection is indeed always in the intersection:
>
> Let $\vy \not \in \Bf \cap \B$. First, we define $\vz$ as the projection of $\vy$ onto $\B$, so $|| \vz- \vx ||  \leq \epsm$. Next, we perform a line search on the segment of endpoints $\vx$ and $\vz$ until we find $\vu$ satisfying $D_\phi(\vu || \vx) \leq \epsilon$. In other words, $\vu = \alpha \vx + (1-\alpha)\vz$ for some $\alpha \in [0,1]$. We have $\vu \in \Bf$ since $D_\phi(\vu || \vx) \leq \epsilon$. We also have $||\vx-\vu|| = ||\vx - \alpha\vx -(1-\alpha) \vz|| = (1-\alpha)||\vx-\vz|| \leq \epsm$ (because $\alpha \in [0,1]$ and $||\vz-\vx|| \leq \epsm$), which means $\vu \in \B$. This concludes the proof that $\vu$, the result of this procedure is always in $\Bf \cap \B$.
>
> > "... it is unclear why Figure 1 is generated when $d=1$"
>
> Figure 1 is only a 2D cartoon meant to give visual intuition of our method so we chose $d=1$ to be easily readable. Choosing $d=0.99999$ for example would have resulted in the same visual.
>
> > "The purpose of introducing $\mu$ as the fraction of Gamma functions in Lemma 1 is not clear."
>
> We use $\mu$ to avoid writing the bulky fraction in Equation. 21 multiple times. The fraction ensures that equality (9) holds. But, as suggested by another reviewer, the lemma can be moved to the appendix.
>
> > "Based on my understanding, each iteration of mirror descent involves one forward pass and one backward pass for the input. It is unclear how the computation cost of Bregman divergence training is determined."
>
> It is not exactly like that: there are two separate phases. First, we train the divergence itself, minimizing the objective in Equation. 10, not involving mirror descent. Then we use the divergence in the inner-loop of the adversarial training (with mirror descent in Algorithm 1).
>
> The overall computational cost is composed of the cost of computing $\phi$, $\Psi$ and $\overline\Psi$. The computation of $\phi(\boldsymbol{x})$ is a forward pass on the convolutional neural network $\phi$. The computation of $\Psi(\boldsymbol{x})$ and $\overline\Psi(\boldsymbol{x})$ are back-propagation passes on the neural networks representing $\phi$ and $\overline\phi$, respectively. These computations are implemented using the highly optimized automatic differentiation engine of PyTorch (mentioned in Sec. 3.1 and Sec 5.).

---

> > ### Comment · Reviewer_brGx · 2023-11-21
> >
> > I appreciate the authors' response. I have a few more follow-up questions about the projection issue. It would be helpful if the authors could provide further explanations.
> >
> > 1. How to ensure the exisitence of $\vu$ satisfying $D_\phi(\vu || \vx) \leq \epsilon$?
> > 2. Your proof only shows $\vu\in\Bf \cap \B$, and does not show it is the projection of $\vz$.
> >
> > I'm still not clear about how is the projection onto set $A$ then set $B$ is equivalent to the projection onto $A\cap B$. Thinkg of a point in 2D and two lines $y=x$ and $y=0$, which intersect at $(0, 0)$. The projection onto intersection is $(0, 0)$ while the projection onto $y=x$ then $y=0$ is not necessarily at $(0, 0)$.

---

> > > ### Author Response · Authors · 2023-11-21
> > >
> > > $\def\vy{\boldsymbol{y}}$
> > > $\def\vx{\boldsymbol{x}}$
> > > $\def\vz{\boldsymbol{z}}$
> > > $\def\vu{\boldsymbol{u}}$
> > >
> > > $\def\epsm{{\epsilon_\text{max}}}$
> > > $\def\Bf{{\mathbb{B}_\phi(\vx, \epsilon)}}$
> > > $\def\B{{\mathbb{B}(\vx, \epsm)}}$
> > >
> > > We should have provided a more intuitive explanation for the projection question. Here is a hopefully improved attempt.
> > >
> > > Indeed, as you say, for two arbitrary sets it would not work. In our setting it does because of the special nature of the sets. Concretely, one set is a Bregman ball, the other is an $L^2$ ball and they have the **same center point.**
> > >
> > > The projection is done by first a projecting into the $L^2$ ball $\B$, which is a division by a scalar. Then, from this point on we do a line search towards the (joint) center point. This entire line segment is thus in $\B$! We pick the first point found that is also in $\Bf$. At least, the (joint) center point will yield a valid result.
> > >
> > > > "Your proof only shows $\vu\in\Bf \cap \B$, and does not show it is the projection of $\vz$."
> > >
> > > Yes, that is all what we need to show, since $\vu$ is the result of our projection procedure used in lines 7-11 of Algorithm 1. This $\vu$ is the result of a fast approximation procedure that always belongs to in intersection $\Bf \cap \B$ and not the result of an exact projection into $\Bf$. This is stated in the paragraph below Algorithm. 1: The projection into the intersection is done as a projection into $\B$ followed by a projection into $\Bf$. Since the latter has no closed-from expression, we approximate it by a binary search over the segment....
> > >
> > >
> > > > "How to ensure the exisitence of $\vu$ satisfying $D_\phi(\vu || \vx) \leq \epsilon$?"
> > >
> > > We search for $\vu$ in the line segment having $\vx$ and $\vz$ as endpoints:
> > >
> > > * Step 0: we start the search from $\vz$. If $D_\phi(\vz || \vx) \leq \epsilon$, then we stop and return $\vu \leftarrow \vz$.
> > > * Step 1: otherwise ($D_\phi(\vz || \vx) > \epsilon$), we move to the middle of the line segment: $\vu \leftarrow \frac{1}{2} \vx + \frac{1}{2} \vz$. If $D_\phi(\vu || \vx) \leq \epsilon$, then we stop and return this $\vu$.
> > > * Step 2: otherwise ($D_\phi(\vu || \vx) > \epsilon$), we move $\vu$ to the middle of the line segment having the old $\vu$ and $\vx$ as endpoints : $\vu \leftarrow \frac{3}{4} \vx + \frac{1}{4} \vz$. If $D_\phi(\vu || \vx) \leq \epsilon$, then we stop and return this $\vu$.
> > > * Step 3: otherwise ($D_\phi(\vu || \vx) > \epsilon$), we move $\vu$ to the middle of the line segment having the old $\vu$ and $\vx$ as endpoints : $\vu \leftarrow \frac{7}{8} \vx + \frac{1}{8} \vz$. If $D_\phi(\vu || \vx) \leq \epsilon$, then we stop and return this $\vu$.
> > >
> > > .....
> > >
> > > So $\vu$ is moving towards $\vx$ starting from $\vz$. In the worst case, $\vu$ ends up very close to $\vx$: $\vu \approx \vx$. Since $\vx$ is in the interior of $\Bf$ (because $D_\phi(\vx || \vx) = 0 < \epsilon$), its neighborhood is also in $\Bf$. In particular, the point $\vu$: $D_\phi(\vu || \vx) \leq \epsilon$.
> > >
> > >
> > > More formally, the value of $\vu$ at a step $i$ follows a series $\vu_i = \alpha_i \vx + (1-\alpha_i) \vz$, with $\alpha_i=1 - (\frac{1}{2})^i$. This series $\{\vu_i\}$ converges to $\vx$ since $\{\alpha_i\}$ converges to 1. We know that $D_\phi(\vx || \vx) = 0 < \epsilon$ so $\vx$ is in the interior of the ball $\Bf \subset \mathbb{R}^n$. Thus, there exists a value of $i=N$, for which $D_\phi(\vu_N || \vx) \leq \epsilon$. The $\alpha$ in the proof above is set to $\alpha=\alpha_N$, and $\vu=\vu_N$. In summary, the existence of $\vu$ satisfying $D_\phi(\vu || \vx) \leq \epsilon$ is ensured by the convergence of $\{\vu_i\}$ to $\vx$ and by the fact that $\vx$ is in the interior of $\Bf$.

---

### Official Review · Reviewer_zXKR · 2023-10-31

**Soundness:** 2 fair
**Presentation:** 3 good
**Contribution:** 3 good
**Rating:** 6
**Confidence:** 3

**Summary:**

The paper investigates the corruption robustness of classifiers, and proposes a variant of adversarial training with perturbations sought according to a similarity measure, which is learned using corruption models. The similarity measure is defined as Bregman divergence, with a learned base function, and mirror descent is employed for seeking perturbations. The experimental results show that the approach outperforms $\ell_2$ and LPIPS based adversarial training towards corruption robustness.

**Strengths:**

- The paper is well written and the overall approach to employ Bregman divergences and mirror descent is well-motivated
- The discussion of the learned base functions for Bregman divergence for different corruptions provides interesting insights towards less artificial threat models.
- The experimental results show good performance on CIFAR-10-C, compared to $\ell_2$ adversarial training and RLAT.

**Weaknesses:**

- The idea of learning similarity measures and generating suitable adversarial examples is interesting. However, since the work aims to increase the robustness of classifiers it needs to be discussed to what extend the corruption model can be practically known.
- The method is compared to $\ell_2$ adversarial and RLAT method in the experiments, but in-distribution performance and data augmentation baselines seem to be missing.

**Questions:**

- Could the authors provide some more information on the in-distribution performance and on the performance of data augmentation baselines (which might also use the corruption model $\tau(x)$)?
- In Table 3 the corruption robustness accuracy is shown for the different models and on different corruptions. Do the authors have any insights on why the model trained on contrast (via learning the similarity measure) performs well on both contrast and fog, whereas the model trained on fog performs worse in both categories?

---

> ### Author Response · Authors · 2023-11-16
>
> Our method presents real-world practical robustness, it approaches the problem from an adversarial learning perspective, and we have included the in-distribution results
>
> > "... since the work aims to increase the robustness of classifiers it needs to be discussed to what extend the corruption model can be practically known."
>
> We show that we can train models that are robust to real-world corruptions like Fog. CIFAR-10-C actually contains computational models for creating corrupted images from clean ones. Otherwise corruptions can be obtained through samples collected in the real world and are thus available in this sense.
>
> > "The method is compared to adversarial and RLAT method in the experiments, but in-distribution performance and data augmentation baselines seem to be missing."
>
> Our method approaches the problem of robustness from an adversarial perspective (as RLAT). Importantly, our method is agnostic to the architecture of the classifier (RLAT is not) and thus can be combined with any of the approaches based on data augmentation, which are thus orthogonal. The in-distribution performance is given in Table 2 and Table 3 under the column "standard accuracy".
>
> > "Do the authors have any insights on why the model trained on contrast (via learning the similarity measure) performs well on both contrast and fog, whereas the model trained on fog performs worse in both categories?"
>
> We had already addressed this question (as good as we could) at the end of the paragraph **Adversarial training** in Sec.5: "One hint is that the three corruptions are similar in nature but this behavior requires further investigation."

---

> > ### Comment · Reviewer_zXKR · 2023-11-22
> >
> > Thanks to the authors for addressing my questions. While I share other reviewer's concerns regarding the limited experiments (incl. data augmentation baselines), I consider the overall approach to be a good contribution. I updated my score.

---

### Official Review · Reviewer_GMTd · 2023-11-02

**Soundness:** 3 good
**Presentation:** 4 excellent
**Contribution:** 3 good
**Rating:** 8
**Confidence:** 4

**Summary:**

This paper introduces an interesting learned similarity metric between images based on Bregman divergence and shows that this metric allows for the training of classifiers more robust against image corruption. L_p distances are widely used to assess the robustness of discriminative models because of mathematical convenience, not because they capture a principled notion of invariance in the data domain. The proposed Bregman divergence is demonstrated to be a promising alternative, contributing both to the literature on metric learning out-of-distribution robustness.

**Strengths:**

* The paper presents a mathematically elegant formulation for metric learning (via self-supervised Bregman divergence learning) and adversarial training (based on mirror descent).
* The authors show preliminary evidence that their method learns a “geometry of image corruptions.”
* From a brief literature review (and bolstered by the paper’s thorough review of related work) I believe the proposed training procedure is indeed novel.
* The comparison to the LPIPS divergence metric shows that the proposed Bregman divergence is an effective metric for learning robust models.
* The methods are very well presented and approachable.

Minor comments:
* I found Table 1 to be very helpful in making the work approachable.
* Algorithm 1 nicely documents the settings for hyperparameters.

**Weaknesses:**

* The experiments are only conducted on CIFAR10-C, which is a single, simplistic dataset. I would like to believe that the results would hold on a more complex dataset like ImageNet-C, but this is not demonstrated in the paper. The Conclusions seem to acknowledge that scaling to larger convex architectures for the base function could be challenging.
* One of the issues of using Bregman divergences to measure image similarity is that these divergences are not necessarily symmetric. Many image augmentations are symmetric.
* Lemma 1 is not important to include in the main text. The alternative approach in A.2 is a much simpler method to draw samples, which does not require the complicated statement of the Lemma; in fact the experiments use the simpler version in A.2! Moreover, due to typicality, sampling from a normal distribution in high dimension will essentially sample from the surface of a hypersphere.
* The computation of the $\Gamma$ function should be tractable, and not require approximation.
* The experiments are not particularly strong. The proposed approach does not perform best on the zoom blur corruption.

Small issues:
* I believe that the paper’s contribution on defining a new method for metric learning should be highlighted in the abstract.
* It feels like Equation 3 is out of place. Perhaps it should be in the next section on Mirror descent? I think it should also be admitted up front that while this projection is unique, it is not available in closed-form for neural networks (Table 1), thus you use a line search heuristic.
* In Table 1, the mirror map for KL divergence is missing a +1 term.
* In the Mirror Descent paragraph in Section 2, I think it should say “mapping $z^{t+1}$ back to the **primal** space”.
* $\tau$ is used before definition in Equation 7 (def isn’t until Section 3.3).
* The caption of Figure 2 should be re-written. I had to read it a couple of times to understand what the plot is showing.

Small typos:
* “settings yield**s**” should be plural in the abstract.
* In the 1st sentence, the phrase “the way” is awkward.
* “AT was found to **improve also**” should be “also improve”.
* In 3.2 you use a \citet when it should be a \citep for Fenchel.
* In the first paragraph of Section 5, Hendrycks should be a \citet.
* Adversarial Training paragraph page 8: “use it”
* Last sentence: “cope” should be “scope”.

**Questions:**

1. Did you assess the L_p adversarial robustness of your models trained with the Bregman notion of robustness? Do you have any idea whether your robust models exhibit any of the beneficial characteristics (e.g. perceptual gradients) observed in robust models trained with adversarial training?
2. When you are training the Inverse Map (Eq. 7), do you use a `detach` operation in practice, i.e. to train the inverse map to emulate the base function without optimizing over the parameters of the base function? As written, this loss would seem to indicate that the base function is optimized to emulate the inverse map.
3. Do you have any metrics of the fit of the inverse map on the test data? Isn’t it also true that the inverse map could be fine-tuned during test time on the test points (i.e. the learned inverse map is a form of amortization).
4. How are corruptions $\tau$ sampled during training? Do you draw a single pair $\tau, d$ for each training image as an augmentation?
5. Since the projection $\Pi_K$ is not available in closed form for a learned base function, the paper uses a binary search heuristic. Can you show that this procedure would lead to convergence in the limit $\eta \rightarrow 0$?
6. Do you have any observations or conjectures about the right modes in the divergence plots (Figure 2)?
7. What is your choice for distance $d$ in Figure 3?

---

> ### Author Response · Authors · 2023-11-16
>
> We would like to thank the reviewer for the feedback ranging from high-level comments to the finer details of the implementation. As you suggest, we will emphasize better that we propose a novel method for metric learning, which is indeed the core contribution of our work with robustness only as one possible application.
>
> > No experiments on ImageNet.
>
> We would have liked to but with the limited GPU compute available to us it would take about one month. We are looking for a compute grant for the next or final version.
>
> > "One of the issues of using Bregman divergences to measure image similarity is that these divergences are not necessarily symmetric. Many image augmentations are symmetric."
>
> Indeed, we have considered the symmetrization $S_\phi$ of the Bregman divergence $D_\phi$, which, however, loses the triangle inequality:
>
> $$
> S_\phi(\boldsymbol{x'}, \boldsymbol{x}) = D_\phi(\boldsymbol{x'} \parallel \boldsymbol{x}) +  D_\phi(\boldsymbol{x} \parallel \boldsymbol{x'})
> $$
> We have substituted $S_\phi$ for $D_\phi$ throughout the approach and we re-run the evaluation in Sec. 5. The symmetry did not provide any improvement compared to the proposed $D_\phi$.
>
> > "Lemma 1 is not important to include in the main text."
>
> You are right. We will change that.
>
> > "Did you assess the L\_p adversarial robustness of your models trained with the Bregman notion of robustness?"
>
> We have not done this experiment.
>
> > " When you are training the Inverse Map (Eq. 7), do you use a `detach` operation in practice"
>
> Exactly! We detach $\Psi(\boldsymbol{x})$ from the computational graph of $\Psi$ to ensure that the optimization step taken to minimize Equation. 7 only modifies the parameter of the inverse map $\overline\Psi$ without affecting the parameters of the map $\Psi$.
>
> > "Do you have any metrics of the fit of the inverse map on the test data?"
>
> The mean square error is minimized down to < 0.001.
>
> > "Isn’t it also true that the inverse map could be fine-tuned during test time on the test points (i.e. the learned inverse map is a form of amortization)."
>
> Indeed, fine-tuning the inverse map during test time could further improve the results. However, the current implementation keeps the inverse map frozen once the training is done.
>
> > "Do you draw a single pair $\tau$ $d$ for each training image as an augmentation?"
>
> Yes, the pair is (uniformly) sampled for each image in the mini batch during each training epoch.
>
> > "Can you show that this procedure would lead to convergence in the limit $\eta \to 0$?"
>
> The convergence of the mirror descent requires the optimized function to be $\beta$\-smooth. In our case, the maximized function is a loss function $l$ over the output of a DNN which is not smooth. So unfortunately, even if the projection $\Pi_K$ were exact, the algorithm would not be guaranteed to converge to the global maximum.
>
> > "Do you have any observations or conjectures about the right modes in the divergence plots (Figure 2)?"
>
> The learned base functions $\phi$ output a scalar for an input image. These scalar values are totally abstract to us as they are leaned though self-supervision. They can be thought of as an "entropy" for images (see Table 1).
>
> > What is your choice for distance in Figure 3?
>
> $d$ is uniformly drawn from the interval $[10^{-7}, 10^{-3}]$.

---

> > ### Comment · Reviewer_GMTd · 2023-11-20
> > **Thank you for your response**
> >
> > I appreciate the detailed answers to all of my questions. I am inclined to keep my score, contingent on the authors clarifying the paper to address my comments and the comments of other reviewers (in particular Reviewer brGx's question about ensuring the projection operation is in the intersection of $B$ and $B_\phi$).
> >
> > I reiterate my assessment that the experiments are not particularly strong, but the paper is interesting, novel, and methodologically sound.

---

### Meta-Review · Area_Chair_zt33 · 2023-12-04

**Metareview:**

The authors propose replacing the $L_p$ ball with a learned Bregman ball for adversarial training.

- **Novelty** of the work is limited; the authors essentially propose replacing the $L_p$ ball with a Bregman ball via convex neural networks. However, in terms of new techniques, tricks, machinery, etc., the paper does not have much to offer. The proposed approach is a combination of heuristic steps (Eq. (7) for training the Fenchel conjugate network, Eq. (10) as the loss for training the main network, the projection step, see below) for solving the problem, without any further intuition and novelty.
- **Motivation and Clarity** can be improved. For instance, it is unclear how $\tau(x)$ is constructed and why making $\tau(x)$ to the original image in terms of the Bregman divergence may necessarily result in learning better distances. I believe the quality of the Bregman will heavily depend on how good an adversarial example $\tau(x)$ is, thus making it a co-dependence issue.
- I agree with Reviewer brGx about the **projection step**. The projection is a crucial step in the proposed method, and it could benefit from more novel methodologies. Additionally, a single projection step is insufficient for convergence (vs. iterative Bregman projections). Also, the line search step is not the correct way of projecting into a Bregman ball.
- **Complexity** of the implementation is quite high. For instance, learning an autoencoder-type architecture by itself is a challenging optimization problem (Eq (7)).

Overall, I believe the paper would benefit from another round of revision and additional clarifications. Also, the novelty of the work is limited in terms of offering new techniques for solving the problem.

**Justification For Why Not Higher Score:**

Please see the comments in the meta review.

**Justification For Why Not Lower Score:**

N/A

---

### Decision · Program_Chairs · 2024-01-16

Reject